# Biomedical Microtechnologies Beyond Scholarly Impact

**DOI:** 10.3390/mi12121471

**Published:** 2021-11-29

**Authors:** Maria Vomero, Giuseppe Schiavone

**Affiliations:** 1BioEE Laboratory, Electrical Engineering Department, Columbia University, New York, NY 10027, USA; mv2803@columbia.edu; 2Research Management & Innovation Directorate, King’s College London, Tower Wing, Guy’s Hospital, London SE1 9RT, UK

**Keywords:** medical devices, bioelectronics, medical engineering, bioMEMS, micro- and nano-fabrication technologies, medical microtechnology, reliability testing, biostability testing, preclinical validation, translational research

## Abstract

The recent tremendous advances in medical technology at the level of academic research have set high expectations for the clinical outcomes they promise to deliver. To the demise of patient hopes, however, the more disruptive and invasive a new technology is, the bigger the gap is separating the conceptualization of a medical device and its adoption into healthcare systems. When technology breakthroughs are reported in the biomedical scientific literature, news focus typically lies on medical implications rather than engineering progress, as the former are of higher appeal to a general readership. While successful therapy and diagnostics are indeed the ultimate goals, it is of equal importance to expose the engineering thinking needed to achieve such results and, critically, identify the challenges that still lie ahead. Here, we would like to provoke thoughts on the following questions, with particular focus on microfabricated medical devices: should research advancing the maturity and reliability of medical technology benefit from higher accessibility and visibility? How can the scientific community encourage and reward academic work on the overshadowed engineering aspects that will facilitate the evolution of laboratory samples into clinical devices?

## 1. Introduction: The Impact Paradox

Micro- and nano-technologies applied in the medical field have long surpassed science fiction in evoking ideas for new treatments, and this is increasingly evident when browsing the latest published research. Scholarly publications that present advances in medical technology typically report first-time demonstrations of new therapeutic or diagnostic capabilities in preclinical or clinical settings. While contributing to rising and maintaining hopes within target patient communities, this also produces a series of secondary effects, as we elaborate hereafter. When publishing medical research that employs a new technology, the research and development aspects related to the technology itself often remain in the background, and sometimes are not thoroughly reported, as they tend to fall into methodology sections or supplementary information. Technology researchers are, therefore, encouraged to chase the medical demonstrations needed for their work to achieve higher visibility and make headlines. The captivating nature of the glistening results obtained in first-time patients or animal models may, however, eclipse important work on technology innovation that is instead essential for the development of clinic-apt medical tools to be used in research and ultimately in patient care. On the one hand, this self-established mechanism requires researchers to conduct preclinical or clinical studies to confer credibility to their technology, and on the other hand this causes the community to overlook the importance of intermediate engineering steps in the development of a medical device, highlighting the lack of standardized processes that guide technologies to the users. In this respect, we believe that defining a clear and comprehensive strategy to advance technology from laboratory demonstrations to tools that can be ultimately employed for medical research is a compelling topic that must be brought forward to the attention of the entire community (Figure 1).

The current order of things comes with the tangible risk of deterring academic researchers from developing robust technologies and optimising their reliability, in favour of presenting novel, more eye-catching results with preclinical and clinical experimentation. This, in turn, causes a considerable rise in the number of scientific publications on new medical devices, that report preliminary experimentation data as the commonly accepted claim to clinical usability, with little focus on the reliability of the underlying technology. As a consequence, the scholarly literature seems to constantly overcome barriers, whereas clinics hosting patients must still rely on “obsolete”, yet at least proven, robust and reliable devices. Paradoxically, the academic social and financial environment compels researchers to produce novelty, rather than establish continuity in their streams and aim for their devices to reach the bedside. As a result, a multitude of pivot studies is fed to and promoted by publishers, which do not build on previous efforts but rather tend to stir direction with favourable winds.

Various factors have contributed to the establishment of the current situation. Scientific publishers regard optimisation and robustness improvements as incremental research, and academic communities tend to consider such efforts as an industry-only prerogative. We believe, however, that this construct should be amended, and on several grounds: patient communities may be left with a sense of false hope when confronted with novelties that produce no follow-up; hype-focused research is more useful for the academics authoring it than for the general public and the organisations funding the research, including taxpayers; the need for a sensible and convincing downstream strategy should be especially encouraged when preclinical research is conducted on animal models, as it is important for the community to see such practices as necessary steps enabling academic studies to evolve into concrete healthcare benefits. In light of the considerations above, we would like our readers to ponder on whether research impact quantification should be adapted to include measures of how, and to what extent, research exerts influence and reaches end users, clinicians and patients in our case, beyond the scholarly metrics.

## 2. The Influence of Microengineering Methods in Medical Technology Research

Micro- and nano-fabrication techniques have heavily influenced advances in the biomedical field by enabling the capability to miniaturize and batch-fabricate complex biosystems and biosensors with flexible designs and multimodal operation. Microfabrication processes were originally developed for the semiconductor industry and not specifically for medical applications. Yet, soft lithography and photolithography, as well as film deposition, etching and focused ion beam lithography, are some of the techniques that allowed the evolution of novel drug delivery systems and implantable, wearable or injectable biosensors. Microfabrication made the range of possible designs for medical devices much wider, and it has the potential to greatly improve the biotic/abiotic interaction via design optimization and device property customization.

Photolithography is one of the most commonly used patterning techniques at the microscale, and uses ultraviolet light to transfer patterns to photosensitive resists in order to create detailed 2D and 3D structures. It was introduced in 1976 and found its primary use in integrated systems and semiconductors [1]. Together with thin film deposition and etching techniques—defined as (1) the application or growth of a material in the form of layers onto a substrate or microstructures and (2) the process of selectively removing some of those materials, respectively [2]—photolithography has created the manufacturing foundation for a multitude of bioelectronic technologies. Only after cleanrooms became well-established in the industry and equipment prices dropped did access to this type of infrastructures become more widely available to research institutes, paving the way for investigators in biomedical areas to make use of microtechnology to their advantage [3]. This marked the rise of bio-microelectromechanical systems (BioMEMS) leveraging microfluidics, optics and multimodal functions [1]. For instance, access to photolithography, in combination with other landmark techniques of microfabrication, finally made it possible to manufacture individual or arrays of micro- and even nano-electrodes, bringing to the world the benefits of low capacitance, high current densities and fast response [3]. Notable examples of medical devices that have emerged in response to subsequent advances in microfabrication and biomaterials, and that have reached clinical experimentation stages, are the retinal prostheses developed in the 1980s [4,5], and the silicon micromachined, intracortical microelectrode arrays, named Utah arrays [6]. Both technologies have been granted approval for clinical experimentation by the US Food and Drugs Administration following decades and hundreds of millions of investments [7,8].

Ever since the spread of semiconductor foundry equipment and processes, significant research in this field has aimed at adapting and applying microfabrication processes to materials and formats that are better suited to the wide spectrum of biological applications. As part of this transformation, polymers have progressively taken the spotlight as candidate structural materials for wearable and implantable devices [9]. The 1990s saw the emergence of micropatterned electrode arrays fabricated on polyimide substrates [10], which were validated and corroborated by pioneering efforts in bringing this new technology into the body [11,12], a research stream that is still ongoing and in full bloom [13]. Similar approaches have been published involving the use of other polymers, such as SU-8 [14] or Parylene-C [15]. In comparison, however, the higher reliability achieved following decades of continued development in polyimide electrode technology have made it possible for researchers to conduct more advanced and elaborate studies aiming at evaluating therapies in primate models [16,17] and humans [18,19,20].

Another significant stream of research aims at applying microfabrication processes to elastomer materials that reduce the device stiffness to suit different wearable and implantable applications. Soft elastomers such as polydimethylsiloxane silicone (PDMS), popularly used as biocompatible molded external shell for implanted devices of different sorts [21,22], have now been shown to be compatible with full integration into wafer-scale microfabrication processes [23]. Examples of devices microfabricated on silicone elastomer include implantable neural interfaces for the central nervous system [24,25] and peripheral nerves [26], as well as wearable electronic sensors that conform the human skin [27]. Other important research directions stem from the inclusion of biodegradable electronic materials in transient implantable devices that are designed to be metabolized once they have served their purpose inside the body [28].

Finally, it is important to note that progress in microfabricated medical devices also benefits from complementary advances in other closely associated areas. Research in microelectronics has led to the development of microchips with increased computational capacity and better energy efficiency, which will power future implantable devices [29,30]. New engineering strategies are being studied to develop adequate packaging techniques for bioelectronic implants using for instance high barrier performance materials [31] or complex architectures [32]. Advances in organic electronics will supply medical devices with better performing biointerfacing materials for charge transduction in sensing and stimulation applications [33,34,35,36,37]. Advanced compact electrical interconnections will play a key role in the practical usability of future high-channel count medical devices [38,39].

Within the multitude of possible approaches for the manufacturing of bioelectronic medical devices, it is undeniable that microfabrication tools and techniques have played a critical role in the advancement of experimental research, both in vitro and in vivo, and that they still hold promises as fundaments of the biotechnologies and medicine of the future. Bringing forward and vulgarizing the technologies that enable new medical devices would be beneficial not only for engineers, but also for the clinical and patient communities. We believe that research on the engineering and technical aspects of medical devices must be able to stand on its own feet in a position of visibility, without the need for the legitimation of fame ensured by preclinical and clinical results. This will provide a more widespread and accessible understanding of the progress, difficulties and potential benefits of new engineering tools, and, hopefully, more engaged interest and concerted contributions to tackling the challenges that prevent experimental devices from reaching advanced research stages. Perhaps researchers working on medical technology could gain the high visibility needed for such a paradigm shift by backing their claims with extensive reliability testing data and by making their results readable by the non-engineering audience that composes their end users. We speculate that, as a result of such transformation process, the subsequent (pre)clinical research stages could benefit from higher reproducibility and chances of success.

## 3. The Challenge of Predicting Clinical Adequacy in Non-Clinical Settings

Translational research is a framework that aims at rendering basic science applicable to clinical settings. However, the poor translatability of preclinical research into clinical applications is consensually recognized by both academia and the industry [40]. In general, the size of the resources invested into biomedical research is not followed by proportionally commensurate advancements in treatments, diagnostics and prevention of human diseases. By 2016, almost 1 million journal papers from clinical trials were published [41], but most of them ended up contributing to the billions of dollars (about 85% of the total spent) wasted annually across medical research [42]. Before approving a medical device for a first-in-man clinical trial, regulatory bodies examine each case based on safety, biocompatibility, function and efficacy evaluations most often extracted from animal tests. The financial and ethical burden associated with such resource intensive validation takes most players out of the game [43], and many promising technologies remain anchored at the stage of proof-of-concept instead of paving the way to clinical practice or even new research opportunities. Therefore, extensive validation activities on biomedical technology have been considered so far by academics almost as an industry prerogative, whereas university laboratories should only shine bright but brief light on promising new ideas.

What makes the biomedical technology field particularly inefficient? Although translational research aims at increasing the chances of success of the clinical/human trials based on laboratory-based discoveries, it is compromised by the misconception that studies on animal models directly enable or justify human clinical studies. However, the claim that a biomedical technology can be translated across species is not always applicable, because it does not take in consideration the different phases, sequential areas of activity, and feedback loops necessary to enable population-level outcomes and truly benefit society [44]. Medical technology development is not a quick and linear process, and it is not financially sustainable by many. Even the most fascinating discoveries can fail to qualify for therapeutic developments because they are irrelevant to human diseases, lack funding or miss the technical expertise to further advance through the translational gap, also known as the “valley of death” [45]. Not being able to accurately predict the outcome of a prospective clinical trial during preclinical and animal studies—and to establish solid safety profiles—leads to a drastic increase in project failures. Nonetheless, basic and applied scientific research typically enjoys rewards almost exclusively based on the scientific output generated, rather than stepwise progress toward reaching clinical application [40]. Despite potentially disastrous financial implications, failure to attain clinical relevance does not significantly affect the academic reputation of biotech inventions, even in cases where they are unlikely to succeed in addressing and solving the real healthcare challenges they target. Data show that many published findings in biomedical research are, in fact, misleading, not accurately reported and cannot be reproduced [41]. We suggest that this may also be a consequence of the fact that scholarly publishers in biomedical technology and the academic community around them do not usually require nor expect elements of reliability testing when considering a manuscript for publication, and rather assess submissions chiefly based on other criteria such as novelty and the extent of in vivo testing.

Additionally, methodological flaws and poor experimental designs in preclinical in vitro and in vivo studies can lead to systematic bias, unreliable results and inaccurate conclusions. These challenges prove difficult to identify and tackle because, despite drawing from the highly regulated fields of pharma and MedTech, researchers in biomedical technology cannot refer to any agreed standards and/or guidelines when preparing an academic research manuscript. The readiness of a new technology for an animal study is not well-defined in academic settings, and it is not clear when and to which extent an in vivo experiment is useful and/or necessary in order to validate a device in the early stages of development. It is though generally assumed that in vivo trials are more valuable than thorough lab-bench test procedures, and research groups are often drawn to prematurely test technologies in animal studies, only to produce data of poor usability and little translational relevance. Ideally, the outcome of an in vivo study should be predictable in vitro, and technical boundary conditions—within which the chances of success are sufficiently high—should be set prior to such trials. By improving the quality of the hypotheses, the conclusions and results of a study can gain credibility. In the current situation, though, the value of a device intended to interface a living system (i.e., an implant or a wearable device) is mainly evaluated by its ability to survive and perform while interacting with the biotic component (an organ or a body). Such interaction, and critically its extensive variability, cannot be treated today as a standardized test condition across users and institutions, and most research devices cannot, therefore, be expected to meet any standard technical requirement relevant to their class of technology.

We think that blurring today’s neat boundaries between academia and the industry (i.e., novel vs. incremental, creativity vs. optimization) could be beneficial for both academic researchers and industrial partners interested in commercializing bedside innovation. Published academic research in technology and engineering for medical devices could for instance include reliability tests that can serve a stepping stone for subsequent verification and validation campaigns. We object that the routine inclusion of data from thorough engineering testing in manuscripts that report novel biomedical technology should be encouraged if not required. By establishing common testing strategies that apply to classes of similar devices, it would also be possible to provide more accurate perspectives for many biomedical technologies.

## 4. From Ideas to Guidelines

When looking at the MedTech industry, fulfilling regulatory compliance requires significant investment in infrastructure, know-how and resource commitment [46,47]. Despite tremendous progress in device engineering, the vast majority of newly published biomedical technologies are today only employed in preclinical studies, and are incompatible with the technical and regulatory requirements of clinical research. As mentioned in the previous paragraph, a new technology that is backed by extensive lab-bench test data prior to in vivo tests would benefit from both higher reliability when approaching animal experimentation and higher credibility when ideas are pitched to fund technology readiness level (TRL)-advancing projects. At the same time, published research could provide a more objective evaluation of the technology and more accurate prediction of its performance. One aspect that may facilitate a shift towards such publishing practices is the development of community-agreed standards for testing different types of medical devices.

A notable example comes from the field of neural interfacing electrodes, in the form of a series of influential publications that have shaped the way most researchers in the field today qualify their electrodes. In 2008, Cogan published a manuscript presenting a collection of techniques typically used for the electrochemical characterization of bioelectrodes in vitro and in vivo [48]. Combining decade-old knowledge and recent insights into specific electrode materials, geometries and applications, this paper still represents today the go-to guide for researchers designing studies that make use of bioelectrodes. Through the years, the community realised the importance of establishing a commonly agreed frame of reference to assess and compare different electrodes, and an increasing number of papers reported data obtained from the characterization techniques described therein. To this effect, we welcome recent efforts aimed at further formalizing the series of techniques proposed by Cogan’s article into guidelines for the characterization of bioelectrodes to be used in preclinical studies [49]. With the emergence of stretchable implantable devices, adapted guidelines have also recently been suggested to suit the new mechanical signature of soft electrodes [50].

In general, once a growing number of researchers publishes their data adhering to agreed yet unofficial guidelines, the peer review process should gradually conform to such newfound standards. This self-regulating mechanism can be thus leveraged to build momentum into a change of practice. Moreover, we believe that this can bring further benefits within the scientific community. If all preclinical studies from a certain area follow community-agreed guidelines on testing and reporting, researchers will be able to conduct cross-study comparisons and benchmarking. Besides preclinical validation results being of easier interpretation, future implications of such a paradigm shift may include the use of big data analyses to identify hidden patterns and enable further discoveries and development.

We should also note that in this context, researchers in microfabricated medical devices are standing on the shoulders of giants, as the microelectronics industry can lend not only designs, materials and processes, but also decade-established quality control and reliability testing routines. Scholarly reporting of statistical process control data on experimental microfabricated medical devices would be, in the authors’ opinion, a valuable contribution to the advancement of this research field and a means to build better cases for regulatory compliance. In general, regulatory compliance is a very broad topic that involves different technical aspects depending on the specific technology and application, as well as different requirements depending on the target geographical area and associated governing regulations. In general, however, we believe that if academics were to collectively develop standardized testing and subsequently adhere to it in their research dissemination, this would pre-emptively improve technology readiness and thus facilitate regulatory compliance. We likewise infer that this could offer potential investors down the road a higher level of confidence in published data.

## 5. The Art of Bridging Business into Science

Beside the scientific community and publishers, it is interesting to consider how funding bodies are approaching the challenge of advancing academic research into healthcare adoption. Traditionally, competitive funding acquisition is won by research groups presenting innovative ideas that promise breakthroughs in their field. When successful research projects come to completion, with demonstrated academic output in the form of scholarly dissemination, reports, intellectual property, etc., their outcomes reach a certain “academic maturity” which can deter some funders from financing follow-up stages, as they cannot be considered as novel. This phenomenon, known as “the valley of death” for medical device technology, can be detrimental when looking at the return on investment of research funding. Effectively, investing capitals in new ideas with disruptive potential is a noble commitment, yet it will not pay off without substantial possibilities for continued financial support. Research must be sustained even when it abandons the spotlight of novelty and moves instead to the various development stages required to achieve scalability, demonstrate commercialisation and adoption strategies, initiate verification and validation campaigns, etc.

If, on one hand, conventional research grants destined to university laboratories fund promising research, how can academics ensure that their innovation has the opportunity of proving really impactful on the long term? An emerging trend in this respect in the medical device academic community is the inclusion of preliminary data or at least aspects of consideration to clinical usability (device handling, interfacing with existing infrastructures and practices) and regulatory compliance [43]. Despite commendable efforts, however, researchers who want to push their inventions towards adoption in a healthcare system most often fall short of options for pursuing such endeavour within their academic seat. In the majority of cases, private funding must be sought to turn the invention into a product, a process that entails building business plans, planning spin-out projects, pitching to investors, etc. This neat separation of the two worlds of science and business often deters researchers from adventuring further in their adoption or commercialisation journey (lack of interest, adequate training, personnel willing to embark in start-up projects, etc.). Can the community envisage a different funding model? Are there opportunities to fund laboratory research once it approaches what is regarded as the academic boundary?

An example that is introducing some wind of change is the increase in translational funds which, in the specific case of medical technologies, aim at advancing the readiness level of a candidate technology, in view of facilitating and accelerating its adoption in a healthcare system. Building a case for support for the commercial and adoption potential of a novel medical technology is a complex endeavour, and one that academics are not necessarily trained to address. For this purpose, research support infrastructures grouping together experts in the different facets of translational research are therefore of paramount importance. A sound translational research proposal builds on successful outcomes of a standard research project and incorporates a diverse range of elements that contribute to its credibility: a detailed target product profile is created and maintained; an intellectual property management strategy is in place, with established background intellectual property (IP) and agreements between stakeholders in place to regulate the foreground IP; such IP is correctly leveraged into a convincing downstream strategy leading to a clear route to commercialization or adoption; the project has been planned in consultation with patient group representatives, with patient feedback loops incorporated throughout the study implementation; the economic impact must be quantified in terms of cost and financial benefits for targeted healthcare providers and markets (Figure 2). This type of projects clearly requires a diverse set of expertise, and universities are setting up dedicated resources and infrastructure or partnering with other institutions that can offer the support needed.

When making decisions on translational research proposals, funders assess the credibility of the downstream strategy and fund the projects that are thought to have the highest chances of progression. For instance, a plan may be in place to increase the appeal of a device to private investors who may put down further funds, or to implement a strategy of licensing a technology to third parties. In fewer words, the answer to the question “What comes after?” must be clear when evaluating a translational research project, in order to build funding bridges and not piers. Time will tell on whether or not this strategy will help convert more laboratory innovations into bedside improvements; however, we welcome such initiatives as a sign that policymakers are looking at ways to address the gap between the results published in the scholarly literature and the healthcare offer to patients.

## 6. Conclusions

We would like to convey the message that research in engineering and validation protocols can benefit from higher visibility in the field of medical technology, as opposed to being formatted specifically for the engineering community. We believe that future discussions around medical technology should closely involve experts from all the stakeholder parties, including engineers that must take on a responsibility role to *(quote)* ‘*assess and thoroughly communicate opportunities and limitations of existing technologies in an understandable way*’ [51]. This could be achieved, for instance, by a more effective dissemination of technical work to lay audiences, aimed at generating wider interest in the engineering challenges around medical technology. In addition, we believe that giving adequate visibility to the technological aspects of medical research will promote more rigorous practices in experimental design and reporting and, ultimately, more reproducible science.

Economic forces push industry-funded clinical research to adopt certain study design and comparators, and to promote familiarity with a new drug or treatment rather than generate knowledge [42]. At the same time, scholarly publishing is a for-profit business relying on advertising, publication and reprint charges. If we consider all these expenses and the fact that for each dollar spent in R&D, less than one dollar of value is returned [52], where is the revenue that keeps the business moving? What are the long-term effects of this model? Furthermore, it is increasingly well known that the impact of research in medical technology goes beyond scholarly citations [53], so should academic institutions drive change and adapt research scoring criteria and the associated metrics?

Although we cannot provide clear-cut answers to the many questions raised by our elaboration, we hope that the considerations we presented herein will help the debate around these topics gain momentum in the coming years, across disciplines and research communities.

In conclusion, we would like the readers to reflect on the following points.

Solid and robust engineering discoveries are as valuable as clinical findings, because they are the tools that will ultimately enable clinical outcomes.The current academic reward system encourages novel biomedical microtechnologies to be experimented in preclinical studies at an early stage. There seems to be more investment and excitement for in vivo trials than in making technologies reliable, and ultimately usable. This leads to the failure of many projects. Reliability and stability testing of more widely celebrated importance may help minimize such risk.Self-regulated mechanisms for the establishment of quality standards are possible and useful in the academic environment, as long as researchers embrace them, comply with them, and contribute to their improvement and updating.A system-level, “build-upon” strategy that aims at creating funding bridges (and not piers) may on the overall be a suitable solution for an improved return on investment in the field of biomedical technology research.

## Figures and Tables

**Figure 1 micromachines-12-01471-f001:**
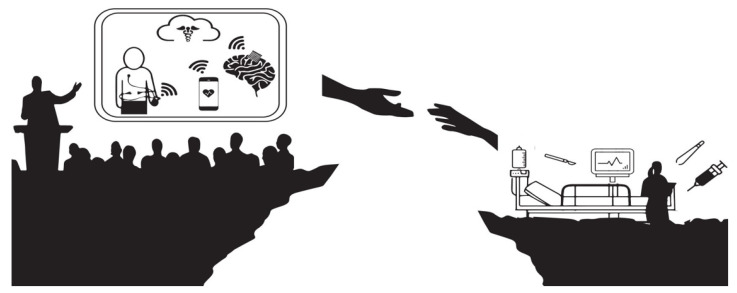
Building the much needed connection between academic research and clinics.

**Figure 2 micromachines-12-01471-f002:**
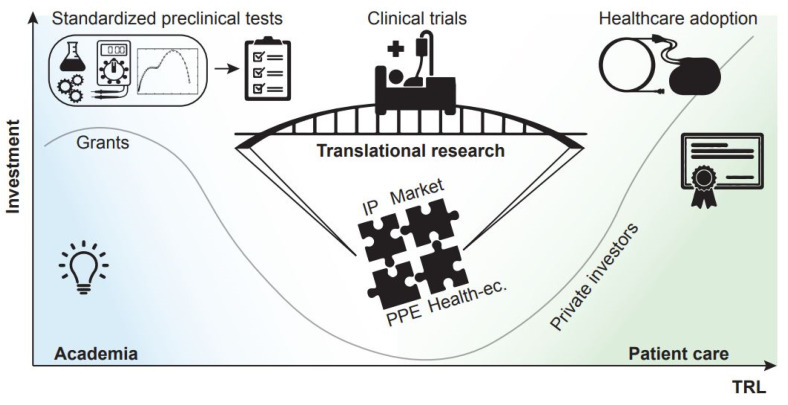
Investment v. Technology Readiness Level (TRL): from academia to patient care.

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
