# Peer review of "Biomedical Microtechnologies Beyond Scholarly Impact"

_micromachines, 2021, doi:10.3390/mi12121471_

Round 1

Reviewer 1 Report

This review by Vomero and Schiavone critically addresses the translation of microengineering methods and devices into the clinic The authors provide a good introduction for a wide readership regarding the different fabrication and uses of microscale technology, continued by an in-depth analysis on the challenges and opportunities in the field, and more importantly provide valuable insight into what can be done to accelerate the actual transition from bench to market. The authors include very good figures that illustrate some of the challenges discussed in the field including how to communicate to understand what problems are worth studying and what drives research and technology development (funding). The article is very well written and structured and provides very useful up-to-date references, thus this is a great access point for researchers of different fields. I support the publication o this work in Micromachines.

Author Response

The authors would like to thank the reviewer for providing a clear summary of the scope of our manuscript and expressing a positive evaluation.

Reviewer 2 Report

Authors –

The authors have put forward a fine effort in presenting their research.

The article is particularly more interesting because of the paradigm shift in the healthcare space because of COVID-19, and this article could be beneficial for researchers interested in the current and future research.  

To maintain the scope and quality of the Journal, there are a few concerns that must be addressed before this article can be accepted for publication.

Below are the comments and suggestions to improve the article:

  1. While the approach is commendable, there is a lack of sound technical premise and evidence. Since the article is about talking about biomedical microtechnologies beyond scholarly impact, regulatory processes, clinical trials, patient adoption, etc. should be discussed. Especially, the regulatory process can have a significant impact to any new technologies, and discussing this will benefit the readers.
  2. An opinion about clinical trials in general, how it has changed post the pandemic, and how researchers can learn from this opinion piece could be important. I would recommend that the authors share their views on how to fast-track clinical trials

Reviewer 3 Report

It is a well-written opinion about microtechnologies and their low impact on the clinic field. The authors mention a wide gap between academic projects and engineering applications. This scenario is overwhelming more impact in countries with moderate capability and without microtechnology infrastructure.

Author Response

The authors would like to thank the reviewer for the positive evaluation and support of our manuscript.